# Association of the rs17782313, rs17773430 and rs34114122 Polymorphisms of/near *MC4R* Gene with Obesity-Related Biomarkers in a Spanish Pediatric Cohort

**DOI:** 10.3390/children10071221

**Published:** 2023-07-14

**Authors:** Joaquín Carrasco-Luna, María Navarro-Solera, Marie Gombert, Vanessa Martín-Carbonell, Álvaro Carrasco-García, Cristina Del Castillo-Villaescusa, Miguel Ángel García-Pérez, Pilar Codoñer-Franch

**Affiliations:** 1Department of Pediatrics, Obstetrics and Gynecology, University of Valencia, 46010 Valencia, Spain; joaquin.carrasco@uv.es (J.C.-L.); marianavarrosolera@gmail.com (M.N.-S.); vanessamartin7891@gmail.com (V.M.-C.); al.carrasco.garcia@gmail.com (Á.C.-G.); 2Department for Biotechnology, Faculty of Experimental Science, Catholic University of Valencia, 46001 Valencia, Spain; 3Biosciences Division, Center for Health Sciences, SRI International, Menlo Park, CA 94025, USA; marie.gombert@sri.com; 4Department of Pediatrics, University Hospital Doctor Peset, Foundation of Promotion of Health, Biomedical Research in the Valencian Region (FISABIO), 46020 Valencia, Spain; crisidc@gmail.com; 5Department of Genetics, Faculty of Biological Sciences, University of Valencia, INCLIVA, 46100 Valencia, Spain; miguel.garcia@uv.es

**Keywords:** haplotype, hyperinsulinemia, melanocortin 4 receptor gene, obesity, type 2 diabetes, hypertension, dyslipidemia

## Abstract

Obesity is a multifactorial disease whose onset and development are shaped by the individual genetic background. The melanocortin 4 receptor gene (*MC4R*) is involved in the regulation of food intake and energy expenditure. Some of the single nucleotide polymorphisms (SNPs) of this gene are related to obesity and metabolic risk factors. The present study was undertaken to assess the relationship between three polymorphism SNPs, namely, rs17782313, rs17773430 and rs34114122, and obesity and metabolic risk factors. One hundred seventy-eight children with obesity aged between 7 and 16 years were studied to determine anthropometric variables and biochemical and inflammatory parameters. Our results highlight that metabolic risk factors, especially alterations in carbohydrate metabolism, were related to rs17782313. The presence of the minor C allele in the three variants (C–C–C) was significantly associated with anthropometric measures indicative of obesity, such as the body mass and fat mass indexes, and increased the values of insulinemia to 21.91 µIU/mL with respect to the wild type values. Our study suggests that the C–C–C haplotype of the SNPs rs17782313, rs17773430 and rs34114122 of the *MC4R* gene potentiates metabolic risk factors at early ages in children with obesity.

## 1. Introduction

Obesity is a multifactorial disease characterized by an unregulated expansion of adipose tissue. The consequence, even in young patients, is the early development of numerous comorbidities associated until recently with ageing, such as cardiovascular diseases, type II diabetes, articular diseases, hepatic steatosis and cancers [1,2,3]. The external factors influencing metabolic regulation have been revolutionized in our modern way of life, causing an increase in obesity cases [4,5,6,7]. It is also clear that these changes affect individuals unequally, and some genes are responsible for this increased tendency to develop obesity in certain individuals [8]. One of them is the melanocortin receptor 4 gene (*MC4R*), coding for a 322-amino-acid-long transmembrane receptor located on one single exon of chromosome 18 in the 18q22 region. This gene is expressed in the hypothalamus, a region involved in appetite control, satiety regulation and energy expenditure [9]. Loss-of-function mutations of the *MC4R* gene lead to increased food intake and a linear increase in lean mass [10]. Moreover, *MC4R* activation has been shown to result in decreased food intake and increased energy expenditure. Single nucleotide polymorphisms (SNPs) mapped to this gene have been identified as predisposing carriers to obesity and related comorbidities [11,12]. The SNP rs17782313 has been shown multiple times to be associated with obesity [13,14,15,16] and with insulin resistance, dyslipidemia and type 2 diabetes risk [16,17,18,19,20]. It has also been identified as a risk factor for obesity in offspring [21,22,23,24,25]. Other SNPs, such as rs17773430, have been associated with a higher percentage of fat mass [26,27,28], and rs34114122 is associated with an increase in plasma ghrelin levels and energy intake [29]. However, with advances in the understanding of the role of the *MC4R* gene, some points remain to be clarified, such as the effect of polymorphism combinations on the gene. Consequently, we conducted the present research in Caucasian children with obesity to perform an association study of individual and multi-locus combinations (haplotypes) of the genetic variants rs17782313 (T>C), rs17773430 (T>C), and rs34114122 (A>C) of the *MC4R* gene (Figure 1) and potential interactions between these polymorphisms and metabolic risk factors. We selected these polymorphisms according to their previously reported association with obesity and because they may have regulatory functions affecting the gene functionality.

## 2. Materials and Methods

### 2.1. Subjects

We have undertaken a cross-sectional and analytical study. The children were consecutively recruited from the outpatient Gastroenterology and Nutrition Clinic of the Department of Pediatrics, Dr. Peset University Hospital of Valencia. They were referred from the pediatrician for the diagnosis of comorbidities (hyperinsulinemia/type 2 diabetes, hypertension, dyslipidemia) in the context of obesity. All the children were of Caucasian Spanish origin. The following criteria were the reason for non-inclusion in the study: eating a specific diet, and carrying a genetic or endocrinological disease and obesity secondary to pharmacological treatment or hypothyroidism. The initial sample was composed of 203 children, and 25 were excluded because they did not complete part of the required data. The definitive sample was composed of 178 children (80 girls and 98 boys), with a mean age of 11.5 years (standard deviation: 2.8). The study was carried out between May and November 2019 after the approval of the local Ethics Committee in accordance with the principles of the Declaration of Helsinki, and informed consent signed by parents or tutors.

### 2.2. Genetic Variables

Genomic DNA extraction was performed using the Qiagen QIAamp DNA kit blood ref. 51106 (Qiagen, Valencia, CA, Spain); 200 µL of blood treated with anticoagulants such as citrate or Ethylenediaminetetraacetic acid (EDTA) yielded 4–12 µg of DNA of high purity. Genotyping and allelic discrimination of rs17782313 (T>C), rs17773430 (T>C), and rs34114122 (A>C) were performed in 3 replicates by quantitative polymerase chain reaction (qPCR) from aliquots of purified DNA with a minimum concentration of 0.2 ng/µL. Predesigned tests by Applied Biosystems TaqMan SNP for Genotyping Assays (Life Technologies, Carlsbad, CA, USA), a 7300 Real-Time PCR System (Applied Biosystems, Warrington, UK) and SDS 1.6.3 software (Applied Biosystems, Warrington, UK) were used following the manufacturer’s instructions.

### 2.3. Clinical Data

Measurements of weight and height were performed following standardized protocols of the International Society for the Advancement of Kinanthropometry. Body mass index (BMI) was calculated as weight in kilograms divided by height in meters squared. BMI z scores were determined following the formula z-score = (X − m)/SD, in which X is the observed value of BMI, and m and SD are the mean and standard deviation value of the distribution corresponding the reference population using the WHO tables. Waist circumference was measured with a flexible non-elastic tape at the belly button of the umbilicus after exhalation. Body composition was analysed by bioelectrical impedance using a Tanita BC-418 MA with 8 contact electrodes (Tanita Europe BV, Hoofddorp, The Netherlands), and the fat mass index (FMI) was calculated as fat mass in kilograms divided by height in meters squared. Systolic and diastolic blood pressure were measured by an M3 Omron digital blood pressure monitor HEM-7200-E8/(V) (Omron Healthcare, Kyoto, Japan), and the average of three measurements was determined. Z score values were calculated on the basis of age, sex and height (https://www.nhlbi.nih.gov/health-topics/fourth-report-on-diagnosis-evaluation-treatment-high-blood-pressure-in-children-and-adolescents, accessed on 19 June 2023).

### 2.4. Laboratory Determinations

Blood obtained after 12-h fast was immediately processed after collection and centrifuged at 3000× *g* and 4 °C for 5 min. Glucose, lipids, apolipoprotein A1 (ApoA1), apolipoprotein B, and gamma-glutamyl transpeptidase were analysed using standard assays (Aeroset System^®^ Abbott Clinical Chemistry Abbott, Wiesbaden, Germany). Insulin was determined using an automated electrochemiluminescence immunoassay (Architect c8000^®^, Abbott Clinic-Chemistry, Abbott Park, IL, USA). Homeostatic model assessment (HOMA) was calculated as (fasting insulin (μU/mL) × fasting glucose (mg/dL)/405). High-sensitivity C-reactive protein and cystatin C were analysed by immunome-phelometry with a Behring 2 (Dade Behring, Marbung, Germany) nephelometer. Adipokines (leptin and adiponectin) and inflammatory markers such as interleukin 6 and tumor necrosis factor-α (TNF-α) were analysed using a Luminex 100 IS (Luminex Corporation, Austin, TX, USA).

### 2.5. Statistical Analysis

The chi-square test was used to assess whether the genotype frequencies fit the Hardy–Weinberg equilibrium compared with the expected frequencies under the assumption of independence. SNPStats software was used to investigate the genotype and haplotype association of *MC4R* gene SNPs (Catalan Institute of Oncology, Barcelona, Spain, available at https://bioinfo.iconcologia.net/SNPstats, accessed on 10 May 2023) using linear regression models. Data are displayed as the mean differences in comparison with the reference category (the most frequent genotype) and 95% confidence interval. The association analysis in this program includes five models of inheritance (dominant, recessive, over-dominant, additive and codominant), and the differences among them lie in the number of copies of the recessive allele required to obtain the effect. The Akaike information criterion (AIC) and Bayesian information criterion (BIC) were employed to select the model that best fits the heritage for each polymorphism. The preferred model is that with the lowest AIC and BIC values. Bonferroni adjustment for multiple comparisons correction with final *p* values of ≤ 0.0167 were considered statistically significant according to the equation: *p* value 0.05/3 = 0.0167 (as we studied three SNPs), and those between 0.05 and 0.0167 were regarded as marginally significant. Bonferroni correction is used to control for type I error when multiple tests of association between genetic markers and a characteristic or disease are performed. All selected *MC4R* gene SNPs before association studies and prior to haplotype analysis were tested for linkage disequilibrium to discard any common segregation. Linkage disequilibrium was tested by the D and D′ coefficients between all pairwise comparisons of loci at the chromosome level. D is the deviation between the expected haplotype frequency under the assumption of no association and the observed frequency. The D’ statistic is equal to D scaled in the (−1, 1) range.

## 3. Results

The main characteristics analyzed in the study population are summarized in Table 1. There were no children with diabetes type 2, hypertension or dyslipidemia.

Genotype, chromosome position, location and allelic frequencies for SNPs rs17782313 (T>C), rs17773430 (T>C), and rs34114122 (A>C) are shown in Table 2. The allelic frequencies met the Hardy–Weinberg equilibrium. Linkage disequilibrium parameters are shown in Table 3.

### 3.1. Polymorphic Analysis

#### 3.1.1. Anthropometric Parameters

The minor allele (C) of rs34114122 is associated with an increase in the BMI z score in an autosomal dominant model. The other two polymorphisms studied did not show an association. No significant association was found between the three polymorphisms tested and FMI (Table 4).

#### 3.1.2. Biochemical Parameters

As reported in Table 5, the minor allele (C) at rs17782313 (T>C) is associated with poorer metabolic characteristics than the most widely represented allele of the same gene. The minor allele is associated with an increase in glucose levels in a recessive model and the HOMA index of insulin resistance in an additive model. The individuals carrying this minor C allele also present significant increases in the levels of cystatin C and leptin, together with a decrease in ApoA1 in recessive models. The minor allele (C) of rs34114122 (A>C) is associated with an increase in the gamma-glutamyl transpeptidase level, although it is within normal levels. In contrast, rs17773430 did not show any association with the metabolic and inflammatory factors studied. In summary, metabolic risk markers are associated mainly with carrying rs17782313 (T>C).

### 3.2. Haplotype Analysis

Table 6 shows the associations between the anthropometric and biochemical traits and the haplotype combinations of the three SNPs selected. Only significant associations are shown.

Regarding the haplotype study, the presence of the minor allele (C) of rs17782313 and the minor allele (C) of rs17773430 together with the minor allele (C) of rs34224122 confers a risk, since it increases the BMI z score average by 2.55 units. This genotypic combination is also associated with an increase in the FMI (5.21 kg/m^2^).

Thus, when three of the minor alleles in the variants of the *MC4R* gene included in the study were combined, a significant association with measures reflecting obesity (BMI and FMI) was found.

The study of haplotype association with biochemical traits showed that the minor allele (C) in rs34114122 combined with the minor allele (C) in rs17773430 and the major allele (T) in rs17782313 slightly increased basal insulinemia (3.28 µIU/mL with respect to the combination of major alleles). However, when the combination of the three minor alleles (C−C−C) occurs, there is a strong increase in insulinemia values (mean of 21.91 µIU/mL with respect to that of wild type), reflecting a state of hyperinsulinemia. Therefore, a stronger effect on altered carbohydrate metabolism is obtained in the case of the combination of the C minor alleles for rs17782313, rs17773430 and rs34114122.

## 4. Discussion

In the present study, we found a clear association between being a carrier of certain alleles of the gene coding for the *MC4R* receptor and metabolic alterations. In this specific case, genetic variants or combinations of genetic variants can affect its normal function, leading to changes in metabolism and increasing the risk of developing related diseases. Our results, in coherence with other studies of the *MC4R* gene, support the relationships between this hypothalamic receptor and the development of impairments classically associated with obesity, even at a very young age [30]. Usually, association studies only allow us to identify a correlation between two events, although in this case the possession of an allele is intrinsic and allows us to infer that the alteration in the gene can modify the function of the protein.

Prior to the haplotype analysis, we determined the Hardy–Weinberg equilibrium as a method for estimating the number of homozygous and heterozygous variant carriers based on allele frequency. Thus, the genetic composition of our study population remains in equilibrium. Additionally, we tested for linkage disequilibrium to eliminate any common segregation. We did not find linkage disequilibrium, indicating independence in the heritability of the studied variants.

The relationships between the hypothalamic receptor gene *MC4R* and the development of disorders classically associated with obesity, even at a very early age, have been described in several studies [29,31,32]. In the present study, we employed a very rigorous test, Bonferroni correction, to determine the p value, which is considered the strongest form of protection against false-positives.

In relation to the anthropometric characteristics, the association analysis revealed weakly significant connections between the parameters analysed and the *MC4R* gene variants. Our results confirm those of other authors who have found an association of rs17782313 with BMI in adults but not in children [30]. BMI is a useful tool for identifying overweight and obesity at the population level, but it does not reflect the amount of adiposity. In this sense, to measure the amount of body fat more accurately, other methods should be employed, such as bioelectrical impedance, which provides the body composition and allows the calculation of several indices indicating an increase in FMI. In this study, it was observed that only the double allele mutated at rs34114122 in the *MC4R* gene was associated with an increase in BMI but not FMI in childhood.

In the case of the biochemical and inflammatory variables, specifically at the rs17782313 locus, the mutation of a T into a C is associated with most of the alterations in the current study, with an increased glucose level in a recessive model and insulin and the HOMA index of insulin resistance in an additive model. Our results regarding carbohydrate metabolism confirm other studies linking this same *MC4R* polymorphism to insulin resistance [28]. Homozygosity for the minor allele of the rs17782313 SNP is also associated with a decrease in ApoA1 and an increase in cystatin C. ApoA1 is a protein that stimulates reverse lipid transport, favoring the elimination of cholesterol, in addition to an anti-inflammatory effect, and thus protecting against atherosclerosis; thus, a decrease in ApoA1 constitutes a risk factor, since it could interfere with its function [33]. Cystatin C is a marker of renal function that has recently been implicated in metabolic syndrome development and can also be a predictive indicator of cardiovascular disease [34]. As reported in Table 4, carriers of the minor allele (C) at rs17782313 (T>C) displayed poorer metabolic characteristics than the children carrying the most represented allele of the same gene. This rs17782313 is mapped 188 kb downstream of *MC4R* gene. We can hypothesize that this mutation could be associated with increased expression levels due to possible hypermethylation involving the development of childhood obesity [35]. We have also found an increase in gamma-glutamyl transpeptidase levels, a marker of liver damage produced in fatty liver disease [36] associated with the minor allele of rs34114122 (A>C). The rs34114122 is located at the 5′ untranslated region of *MC4R.* Non-coding regions may have regulatory functions relevant in disease pathogenesis. In the case of rs17773430 (T>C), no significant differences were found between the mutation and metabolic traits. The rs17773430 (T>C) has no effect on the final protein of the M4CR, but is intragenic for the uncharacterized transcript LOC105372155 from a long non-coding RNA that could be regulating other genes’ expression. Even though these *MC4R* gene polymorphisms are related to characteristics of obesity [37], it is compelling how the various SNPs seem to have different impacts on the carrier’s metabolism. Additionally, it is important to take into account the interaction of other important factors, such as diet, with gene variants when assessing metabolic risk [38,39,40].

With respect to adipokines, rs17782313 (T>C) also correlates with increased levels of circulating leptin. Leptin signaling modulates energy balance through a combination of melanocortin-dependent and melanocortin-independent pathways. These hypothalamic pathways interact with other brain centres to coordinate energy intake and expenditure [40]. Leptin is physiologically secreted by adipocytes in response to sufficient adipose tissue, which causes a decrease in food intake as a feedback mechanism that promotes satiety. However, leptin levels have been shown to increase in obesity, where patients develop resistance to this hormone. Thus, despite high circulating levels of leptin, people with obesity have a decreased sensation of satiety, promoting overweight and obesity. Magno et al. found that patients with the *MC4R* rs17782313 polymorphism had a higher prevalence of severe binge eating with at least one risk allele [41]. The *MC4R* rs17782313 polymorphism may influence feelings of hunger and satiety. Leptin has also been associated with cardiometabolic risk factors and has been proposed as a marker of metabolic syndrome, highlighting the importance of this factor [42]. It has been proposed that when *MC4R* is not synthesized or its functionality is deficient, it does not generate appetite inhibition. The association of rs17782313 (T>C) with leptin supports this hypothesis.

The other polymorphisms studied showed only weakly significant associations with biochemical parameters. Consequently, the results of the association of individual polymorphisms alone could not be of special relevance.

However, when analyzing combinations of these polymorphisms in the study of haplotypes, it was observed that certain specific combinations of polymorphisms are related to variables indicating obesity. In this sense, we found that the combination of the C-C-C minor alleles at variants rs1778782313, rs17773430, and rs34224122 is associated with both BMI and FMI. Specifically, the presence of the minor C allele at the three polymorphisms is associated with an average increase of 2.55 units in BMI and an average increase of 5.21 kg/m^2^ in FMI compared to those in the presence of the three major alleles.

It was also found that the combination of the C allele at rs17782313 and rs17773430 was associated with a more pronounced increase in insulin levels compared to the combination of nonmutated alleles and each of these polymorphisms separately.

Furthermore, we found that the mutated C allele in rs17773430, combined with rs17782313 (C) and rs34224122 (C), induces a higher increase in insulin levels than rs17782313 (C) alone or any other haplotype combination. Our study reveals the importance of haplotype studies and approaches that analyze more than one SNP at a time to evaluate genetic risk. This finding also suggests that this genetic combination may contribute to an increased risk of obesity, body fat accumulation and insulin resistance, which can contribute to type 2 diabetes development. These results support other studies that have also observed the influence of the minor allele on the polymorphisms studied [39,41]. Although it is clear that *MC4R* gene polymorphisms are related to obesity characteristics, it is interesting to note how different SNPs may have different effects on the metabolism of individuals. None of our polymorphisms is a missense variant that could alter the structure/function of the MC4R protein and explain the association described in this work with obesity biomarkers. However, apart from the fact that these SNPs may be in linkage disequilibrium with causative genetic variants, the SNPs themselves may be functional by altering gene expression of *MC4R* or other genes in the region. In fact, the SNP rs17782313 is an eQTL of the *MC4R* gene (https://www.gtexportal.org/home/gene/MC4R, accessed on 20 June 2023), with the C allele being associated with greater gene expression.

These associations between metabolic biomarkers and SNPs can also be explained by the regulatory functions of non-coding areas in/close to a gene, which can affect protein levels without affecting the structure, hence more relevant in this context. Indeed, protein folding and 3D structure are more relevant in case of SNPs in coding regions. These factors could affect the affinity for ligands or messengers. As a result, appetite control is affected, which may increase the tendency to binge eat and, thus, obesity.

We know that a limitation of the current study is the sample size, which may make it preliminary. Larger samples are needed to estimate genetic effects. However, we have identified significant associations after the Bonferroni correction, which added strength to the results obtained.

## 5. Conclusions

In summary, our findings indicate that carrying the minor C allele in the combination of the genetic variants rs17782313, rs17773430, and rs34114122 of the *MC4R* gene is associated with an increased risk of obesity and carbohydrate metabolism alteration. Therefore, the *MC4R* gene and its polymorphisms appear to be an important factor to consider in the evaluation of genetic predisposition to obesity and comorbidities, even in childhood.

## Figures and Tables

**Figure 1 children-10-01221-f001:**
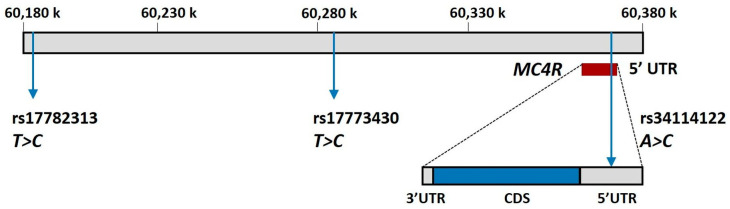
SNP location and chromosome position for SNPs rs17782313 (T>C), rs17773430 (T>C), and rs34114122 (A>C). *MC4R* gene is located between the position 60,371,062b and 60,372,775b at chromosome 18. The rs17782313 is an intergenic variant, located 188 kb downstream of the *MC4R* gene.

**Table 1 children-10-01221-t001:** Demographic, anthropometric and clinical data of the 178 children participating in the study.

Anthropometric and Clinical Parameters		Biochemical Parameters	
Male/Female (%)	53/47	Glucose (mg/dL)	93 ± 7
Age (years)	11.5 ± 2.8	Insulin (µIU/mL)	14.8 ± 8.5
Tanner scale of puberty	2.7 ± 1.2	Homeostatic model assessment	3.6 ± 2.1
Body mass index (kg/m^2^)	28.6 ± 4.2	HDL colesterol (mg/dL)	45 ± 9
Body mass index z-score	3.9 ± 1.6	Triglicéridos (mg/dL)	91 ± 44
Fat mass (%)	35.1 ± 7.0	Apolipoprotein A1 (mg/dL)	131 ± 27
Fat mass index (kg/m^2^)	9.6 ± 3.9	Cystatin C (mg/L)	0.79 ± 0.46
Waist circumference (cm)	95.8 ± 16.6	GGT (U/L)	16.8 ± 7.5
Waist circumference z-score	3.1 ± 2.2	Leptin (ng/mL)	49.3 ± 23.2
Systolic blood pressure (mm Hg)	117 ± 13	Interleukin 6 (pg/mL)	1.9 ± 1.8
Systolic blood pressure z-score	0.62 ± 0.49	TNF-α (mg/mL)	3.5 ± 1.5
Diastolic blood pressure (mm Hg)	68 ± 10	hs-CRP	3.7 ± 6.3
Diastolic blood pressure z-score	0.25 ± 0.44		

GGT, gamma glutamyl transpeptidase; hs-CRP, high-sensitivity C-reactive protein; TNF-α, tumor necrosis factor alpha. Values are expressed with the mean ± standard deviation.

**Table 2 children-10-01221-t002:** MC4R gene SNP description.

SNPdb	Chromosome Position	Location	Major Allele	Minor Allele	MAF	*p*-HWE	eQTL
rs17782313	Chr.18: 60183864	Intergenic	T	C	0.7	0.076	*MC4R*
rs17773430	Chr.18: 60295884	Intragenic *	T	C	0.72	0.85	RP11-396N11.1 RP11-866E20.3
rs34114122	Chr.18: 60372527	5′-UTR	A	C	0.93	0.56	--

MAF, major allele frequency at our population; p-HWE, Hardy Weinberg *p* value (>0.05 assumption of independence); eQTL expression quantitative trait loci, genes whose gene expression is associated with the allelic variation of the SNP in question. Source GTEX Portal. * Regarding the uncharacterized long non-coding RNA transcript LOC105372155.

**Table 3 children-10-01221-t003:** Linkage disequilibrium.

	rs17782313–rs17773430	rs17782313–rs34114122	rs34114122–rs17773430
D	0.0528	0.0335	0.0050
D′	0.5963	0.2689	0.0989

D, deviation between the expected haplotype frequency and the observed frequency; D′, value of D normalized in a range (−1 to1).

**Table 4 children-10-01221-t004:** Associations found between *MC4R* gene SNPs and anthropometric variables.

	rs17782313 (T>C) *n* = 178				rs17773430 (T>C) *n* = 178				rs34114122 (A>C) *n* = 178			
	IM	GT	Mean Dfference * (95% CI)	*p*	IM	GT	Mean Difference * (95% CI)	*p*	IM	GT	Mean Difference * (95% CI)	*p*
BMI z-score	AD	C/C	0.58 (0.08 to 1.07)	0.025	AD	C/C	0.56 (0.08 to 1.03)	0.023	AD	C/C	1.00 (0.21 to 1.79)	**0.014**
FMI (kg/m^2^)	AD	C/C	1.11 (0.05 to 2.16)	0.041	DO	C/C	1.53 (0.25 to 2.81)	0.020	DO	T/CC/C	1.95 (0.06 to 3.84)	0.044

BMI, body mass index; CI, confidence interval; FMI, fat mass index; GT, genotype; IM, inheritage model; SBP, systolic blood pressure. * In comparison with the reference category (the most frequent genotype).

**Table 5 children-10-01221-t005:** Associations between *MCR4* gene SNPs and clinical, biochemical and inflammatory variables.

	rs17782313 (T>C) *n* = 178				rs17773430 (T>C) *n* = 178				rs34114122 (A>C) *n* = 178			
	IM	GT	Mean Difference * (95% CI)	*p*	IM	GT	Mean Difference * (95% CI)	*p*	IM	GT	Mean Difference * (95% CI)	*p*
SBP z-score	RE	C/C	0.49 (0.02 to 0.97)	0.044	RE	C/C	0.49 (0.02 to 0.97)	0.044	SD	A/C	0.15 (−0.22 to 0.52)	0.42
Glucose (mg/dL)	RE	C/C	6.36 (2.22 to 10.50)	**0.003**	SD	T/C	0.46 (−1.64 to 2.56)	0.67	SD	A/C	1.17 (−2.00 to 4.33)	0.47
Insulin (µIU/mL)	AD	C/C	2.53 (0.66 to 4.41)	**0.009**	DO	T/C C/C	2.64 (0.33 to 4.94)	0.026	AD	C/C	2.64 (−0.37 to 5.66)	0.087
HOMA	AD	C/C	0.68 (0.23 to 1.14)	**0.004**	DO	T/C C/C	0.61 (−0.03 to 0.59)	0.035	AD	C/C	0.65 (−0.08 to 1.39)	0.084
HDL-C (mg/dL)	AD	C/C	−1.67 (−4.19 to 0.84)	0.19	AD	C/C	0.09 (−2.32 to 2.51)	0.94	SD	A/C	−5.11 (−9.76 to −0.46)	0.033
TG (mg/dL)	RE	C/C	14.1 (−9.2 to 37.4)	0.24	DO	T/C C/C	7.6 (−3.8 to 19.0)	0.20	SD	A/C	18.2 (1.0 to 35.5)	0.040
ApoA1 (mg/dL)	RE	C/C	−21.8 (−36.4 to −7.2)	**0.004**	AD	C/C	−1.7 (−7.4 to 4.0)	0.56	SD	A/C	−5.9 (−17.0 to 5.3)	0.30
Cys-C (mg/L)	RE	C/C	0.36 (0.16 to 0.57)	**0.000**	DO	T/C C/C	−0.05 (−0.16 to 0.06)	0.35	AD	C/C	−0.07 (−0.20 to 0.07)	0.34
GGT (U/L)	RE	C/C	0.35 (−3.51 to 4.21)	0.86	RE	C/C	−1.91 (−5.36 to 1.54)	0.28	RE	C/C	3.97 (1.15 to 6.79)	**0.006**
Leptin (ng/mL)	RE	C/C	27.8 (10.7 to 45.0)	**0.002**	SD	T/C	6.4 (−2.6 to 15.4)	0.16	DO	A/CC/C	11.32 (−1.03 to 23.67)	0.074
IL 6 (pg/mL)	RE	C/C	0.54 (−0.29 to 1.37)	0.20	RE	C/C	0.81 (0.05 to 1.57)	0.038	AD	C/C	−0.26 (−0.78 to 0.25)	0.32
TNF-α (mg/mL)	RE	C/C	0.48 (−0.40 to 1.36)	0.29	AD	C/C	0.03 (−0.32 to 0.37)	0.89	SD	A/C	0.72 (0.08 to 1.35)	0.029
hs-CRP (mg/L)	SD	T/C	1.00 (−2.67 to 0.67)	0.24	DO	T/C C/C	1.47 (−0.22 to 3.16)	0.09	RE	C/C	4.90 (−3.02 to 12.81)	0.23

AD, additive; ApoA1, apolipoprotein A1; CI, confidence interval; Cys-C, cystatin C; DO, dominant; GGT, gamma glutamyl transpeptidase; GT, genotype; HDL-C, high-density lipoprotein cholesterol; HOMA, homeostatic model assessment; hs-CRP, high-sensitivity C-reactive protein; IL, interleukin; IM, inheritance model; RE, recessive; SBP, systolic blood pressure; SD, over-dominant; TNF-α, tumor necrosis factor alpha. * In comparison with the reference category (the most frequent genotype).

**Table 6 children-10-01221-t006:** Haplotype analysis (*n* = 178 children).

	17782313 (T>C)	17773430 (T>C)	34224122 (A>C)	F	Difference (95% CI)	*p*
BMI z-score	T	T	A	0.5433	NA	-
	T	C	A	0.1375	0.04 (−0.65 to 0.73)	0.91
	C	T	A	0.1359	−0.02 (−0.74 to 0.7)	0.96
	C	C	A	0.1130	0.84 (0.11 to 1.57)	0.024
	C	T	C	0.0283	0.67 (−0.94 to 2.28)	0.42
	C	C	C	0.0261	2.55 (0.96 to 4.14)	**0.002**
	T	T	C	0.0158	−0.35 (−2.53 to 1.83)	0.75
FMI (kg/m^2^)	T	T	A	0.5421	NA	
	T	C	A	0.1386	−0.03 (−1.49 to 1.43)	0.97
	C	T	A	0.1358	−0.21 (−1.73 to 1.3)	0.78
	C	C	A	0.1133	1.81 (0.28 to 3.34)	0.021
	C	T	C	0.0293	0.86 (−2.27 to 4)	0.59
	C	C	C	0.0249	5.21 (1.55 to 8.86)	**0.006**
	T	T	C	0.0160	−1.22 (−5.59 to 3.14)	0.58
Insulin (µIU/mL)	T	T	A	0.5396	NA	
	T	T	C	0.1381	−0.17 (−2.8 to 2.45)	0.90
	T	C	A	0.1305	0.76 (−2.03 to 3.55)	0.59
	T	C	C	0.1215	3.28 (0.68 to 5.89)	**0.014**
	C	C	A	0.0408	0.3 (−4.01 to 4.61)	0.89
	C	T	T	0.0119	−2.48 (−10.26 to 5.31)	0.53
	C	C	C	0.0105	21.9 (13.2 to 30.6)	**0.000**

BMI, body mass index; CI, confidence interval; FMI, fat mass index. NA, non-applicable.

## Data Availability

Data is available when required.

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
