# Peer review of "Association of the rs17782313, rs17773430 and rs34114122 Polymorphisms of/near MC4R Gene with Obesity-Related Biomarkers in a Spanish Pediatric Cohort"

_children, 2023, doi:10.3390/children10071221_

Round 1
Reviewer 1 Report
Authors reported interesting and well written study “Association of the rs17782313, rs17773430 and rs34114122 polymorphisms of MC4R gene with obesity comorbidities in a Spanish paediatric cohort”. However, there are a few points which must be clarified in your manuscript:
1. We advise you to include a table with basic data about the subjects (age, body weight, height, gender, etc.)
2. Furthermore, it would be interested to see how many subjects meet the criteria of metabolic syndrome (see paper https://www.mdpi.com/2227-9067/9/2/204) and the possible association of MS with the mentioned polymorphisms of MC4R gene.
Author Response
Reviewer #1:
We thank the reviewer for the valuable comments.
- There is not abstract, the authors must add this section.
Response: I am sorry. There was a mistake. It is already in the manuscript
- In the keywords section, it is necessary to report more comorbidities associated with MC4R polymorphisms.
Response: We have modified it as follows: type 2 diabetes, hypertension, dyslipidemia
- I recommend the next good references (PMID: 34769060, PMID: 33809061, and PMID: 36612019) for the introduction section of this interesting manuscript. For example, these references for the next phrase: “…………… such as cardiovascular diseases, type II diabetes, articular diseases, hepatic steatosis and cancers [1–3].”
Response: We have added the suggested references [1–3].
- In the following phrase: “The SNP rs17782313 has been shown multiple times to be associated with obesity [13–16].” The reference 14 does not contain the appropriate information for this sentence. Please add the appropriate reference.
Response: We have changed the reference according to your suggestion.
- In materials and methods section, specifically in subjects, please add the comorbidities analyzed. The authors should also add the percentage of children’s sex (female and male), including the study time period.
Response: It was included (Page 2, line 78-79, line 84 and lines 85-86.)
- In the genetic variables, please add the machine name for genotyping, including the number of replicates for the genotypes.
Response: It was added. (Page 3, lines 92, 94, 95, 97 and 98)
- The authors should add the information related to body mass index z-score calculation.
Response: It was included (Page 3, lines 104-106)
- Why was the waist circumference not determined?
Response: Waist circumference was measured (Page 3, line 107-108). We did not include it in the original version because it was a variable not significant. In this revised version it is also shown in Table 1 (Page 4).
- Please define the acronyms when they are used for the first time (e.g., BMI, FMI, and etcetera).
Response: It was done
- -In the table 4, there are several acronyms that should be defined correctly; for example, hs-CRP, C-reactive protein ultrasensible; TNF, tumor necrosis factor alpha; C, confidence interval. In addition, some acronyms definition are need. The significant values should be in bold.
Response: It was done
- -Please add the n for each MC4R polymorphism analyzed in tables, respectively.
Response: All polymorphisms were analyzed in the 178 children. We added the n in the tables.
- The authors should add a new table with anthropometric data (BMI, sex, age), glucose, insulin, leptin, TNF-α, etcetera, including percentages of comorbidities associates with obesity. The values expressed in mean values ± SD.
Response: Table 1 (Page 4) was added with the data required. It is referred in the text (Page 4, lines154-155)
- Please define correctly BMI because the following phrase is incorrect: “BMI is calculated using height, and weight and is a useful tool for identifying……..”
Response: We have deleted the calculation in the discussion section because it is included in the material and methods section (Page 3, lines98-99) in order to improve the clarity of the sentence
- Please add the limitations of this study.
Response: It is added (Page 9, lines 333-336)

Reviewer 2 Report
Here are my comments on the manuscript by Carrasco-Luna et al. entitled “Association of the rs17782313, rs17773430 and rs34114122 Polymorphisms of MC4R Gene with Obesity Comorbidities in a Spanish Paediatric Cohort”
-There is not abstract, the authors must add this section.
-In the keywords section, it is necessary to report more comorbidities associated with MC4R polymorphisms.
-There are some typos in the manuscript.
-I recommend the next good references (PMID: 34769060, PMID: 33809061, and PMID: 36612019) for the introduction section of this interesting manuscript. For example, these references for the next phrase: “…………….such as cardiovascular diseases, type II diabetes, articular diseases, hepatic steatosis and cancers [1–3].”
-In the following phrase: “The SNP rs17782313 has been shown multiple times to be associated with obesity [13–16].” The reference 14 does not contain the appropriate information for this sentence. Please add the appropriate reference.
-In materials and methods section, specifically in subjects, please add the comorbidities analyzed. The authors should also add the percentage of children’s sex (female and male), including the study time period.
- In the genetic variables, please add the machine name for genotyping, including the number of replicates for the genotypes.
-The authors should add the information related to body mass index z-score calculation.
- Why was the waist circumference not determined?
-Please define the acronyms when they are used for the first time (e.g., BMI, FMI, and etcetera).
-In the table 4, there are several acronyms that should be defined correctly; for example, hs-CRP, C-reactive protein ultrasensible; TNF, tumor necrosis factor alpha; C, confidence interval. In addition, some acronyms definition are need. The significant values should be in bold.
-Please add the n for each MC4R polymorphism analyzed in tables, respectively.
-The authors should add a new table with anthropometric data (BMI, sex, age), glucose, insulin, leptin, TNF-α, etcetera, including percentages of comorbidities associates with obesity. The values expressed in mean values ± SD.
- Please define correctly BMI because the following phrase is incorrect: “BMI is calculated using height, and weight and is a useful tool for identifying……..”
-Please add the limitations of this study.
There are only some typos.
Author Response
Reviewer #2:
Thank you for your kind comments about our work. They have helped us to review the manuscript.
Most of the confusions can be fixed if authors explained the location of these mutations and their functional significance in the introduction, (rs17782313 is a Intergenic variant located 188 kb downstream the MC4R gene, rs34114122 is located at 5′ untranslated region of MC4R, rs17773430?)
Response: In order to clarity we have included a Figure (Figure 1), (Page 2).with the SNP location, and we have modified the Table 2 (Page 4-5)
Major comments;
- Abstract is missing in the manuscript.
Response: We apologize the error. In the revised version it is included.
- In the discussion there are several sentences mentioning facts about biomarkers without citing references. Please screen and cite relevant literature.
E.g. Line 223; cite relevant references for ApoA1 and Cystatin C in the sentence “ in ApoA1 constitutes a risk factor since it could interfere with its function. Cystatin C is a….. you may cite https://doi.org/10.1177/0300060520986311.
250-252; Leptin has also been associated with cardiometabolic risk factors and has been proposed as a marker of metabolic syndrome.
Response: According to your suggestion we have included the references proposed, screened the discussion section and cited relevant literature. The changes are highlighted in red in the text and in the References section.
- Most of the parameters you assessed are actually clinical/anthropometric/metabolic biomarkers but not comorbidities. Examples of comorbidities are presence of diabetes, hypertension, dyslipidemia, for which you should have used diagnostic criteria and used different statistical approach to compare subjects with disease vs. d w/o disease.
Based on your conclusions I would suggest modifying the topic “Association of the rs17782313, rs17773430 and rs34114122 Polymorphisms of/near MC4R Gene with Obesity-related Biomarkers in a Spanish Paediatric Cohort”.
Response: It is true. Moreover, in children evaluation we have not found comorbidities according to international criteria but biomarkers that could involve a risk for future development of comorbidities. We have consequently modified the tittle of the Article according to your suggestion.
- Line 223-224; “It could be argued that the latter mutation could be silent and simply have no effect on the final protein or have poorer expression levels.” This sentence is anecdotal. rs34114122 is located in the 5′ untranslated region of MC4R. Therefore, anyway these variants are unlikely mutate the protein (no introns). As a matter of fact, it’s important to remember that non coding regions may have regulatory functions relevant in disease pathogenesis.
Response: We have deleted the sentence and referred that genetic variation in the 5'UTR region may have consequences in the regulation of gene expression. In this way it has been modified in the text (Page 8, lines 266, 271-272, 274-276). The SNP rs17773430 is intragenic but from the uncharacterized transcript LOC105372155 of a regulatory RNA. This fact has been included in Table 2
Minor comments.
- Line 36-37; authors state “Single nucleotide polymorphisms (SNPs) of this gene have been identified to predispose carriers to obesity and related comorbidities” however all the SNPs are not within the gene. For instance, rs17782313 is an intergenic variant, located 188 kb downstream the MC4R gene. I would paraphrase “Single nucleotide polymorphisms (SNPs) mapped to this gene”/ “Single nucleotide polymorphisms (SNPs) located in/near the MC4R”
Response: We have modified the sentence according to your suggestion “Single nucleotide polymorphisms (SNPs) mapped to this gene” (Page 2, line 53-54)
- Line 50; authors state “located in different areas of the gene”. Not all are located in the gene. (rs17782313 is an intergenic variant, located 188 kb downstream the MC4R gene). I would give a summary of the SNPs, location and the functional significance to give a clear idea to the readers.
Response: In order to clarity we have included a Figure with the SNP location (Figure 1) (Page 2) and we have modified the Table 2 with the SNP functionality eQTL(expression quantitative trait loci ) showing genes whose expression is associated with the allelic variation of the SNP (Page 4-5).
- Do you have any reference to support “they are located in different areas of the gene where essential transcription promoter elements are located?”. What about the downstream variant rs17782313? It would be more appropriate to say they may have regulatory functions.
Response: We have changed the sentence to: “We selected these polymorphisms according to their previously reported association with obesity and because they may have regulatory functions affecting the gene functionality” (Page 2, lines 66-68), since it offers a better idea about their possible functional role on gene expression (expression quantitative trait loci, eQTLs) in mechanisms linking them to the phenotype.
- Line 65; sentence looks incomplete. “in accordance with the principles of the Declarationof Helsinki, and signed informed….?”
Response: We apologize the typographic error. The sentence has been completed to: “in accordance with the principles of the Declaration of Helsinki, and the informed consent signed by parents or tutors.” (Page 3, lines 87-88)
- 12-hour fasting is not relevant for genetic studies, but for lipid biomarkers. Delete 12 hour from genetic and mention under, glucose, lipids, apolipoprotein A1 (ApoA1), apolipoprotein B…..
Response: It was done according to your suggestion. The text is at Page 3, lines 117-118.
- What is the reference chart used for blood pressure Z-score determination?
Response: https://www.nhlbi.nih.gov/health-topics/fourth-report-on-diagnosis-evaluation-treatment-high-blood-pressure-in-children-and-adolescents. It was added in the text (Page 3, lines 114-115).
- Systolic blood pressure in not a biochemical parameter. I think it fits better with anthropometric biomarkers after changing the subtitle “Anthropometric and clinical parameters”.
Response: Blood pressure is a clinical parameter and it is considered an important cardiometabolic risk factor. We think that it is more coherent to maintain it with the biochemical and inflammatory parameters.
- In table 4; It’s good to indicted significant p-values boldface.
Response: It is done
- In the table 4 caption; define abbreviations; Cys-oC and ApoA1
Response: It is done
- line 280-281; these associations between metabolic biomarkers and SNPs can also be explained by the regulatory functions of non-coding areas in/close to a gene, which can affect protein levels without affecting the structure, hence more relevant in this context. Protein folding, and 3D structure are more relevant in case of SNPs in coding regions.
Response: We agree. We have changed the text accordingly. (Page 9, lines 320-331)

Reviewer 3 Report
This study explores important correlations between MC4R related SNPs and selected anthropometric, clinical and biochemical markers. Findings are interesting. Discussions can be improved with more insights into the functional significance of the SNPs. With these fixes the paper will be well-received by the readers.
Most of the confusions can be fixed if authors explained the location of these mutations and their functional significance in the introduction, (rs17782313 is a Intergenic variant located 188 kb downstream the MC4R gene, rs34114122 is located at 5′ untranslated region of MC4R, rs17773430? )
My major comments;
1) Abstract is missing in the manuscript.
2) In the discussion there are several sentences mentioning facts about biomarkers without citing references. Please screen and cite relevant literature.
E.g. Line 223; cite relevant references for ApoA1 and Cystatin C in the sentence “ in ApoA1 constitutes a risk factor since it could interfere with its function. Cystatin C is a….. you may cite https://doi.org/10.1177/0300060520986311
250-252; Leptin has also been associated with cardiometabolic risk factors and has been proposed as a marker of metabolic syndrome
3) Most of the parameters you assessed are actually clinical/anthropometric/metabolic biomarkers but not comorbidities. Examples of comorbidities are presence of diabetes, hypertension, dyslipidemia, for which you should have used diagnostic criteria and used different statistical approach to compare subjects with disease vs. d w/o disease.
Based on your conclusions I would suggest modifying the topic “Association of the rs17782313, rs17773430 and rs34114122 Polymorphisms of/near MC4R Gene with Obesity-related Biomarkers in a Spanish Paediatric Cohort”.
4) Line 223-224; “It could be argued that the latter mutation could be silent and simply have no effect on the final protein or have poorer expression levels.” This sentence is anecdotal. rs34114122 is located in the 5′ untranslated region of MC4R. Therefore, anyway these variants are unlikely mutate the protein (no introns). As a matter of fact, it’s important to remember that non coding regions may have regulatory functions relevant in disease pathogenesis.
Consult the flowing paper; Mining the Unknown: Assigning Function to Noncoding Single Nucleotide Polymorphisms https://doi.org/10.1016/j.tig.2016.10.008
Minor comments.
5) Line 36-37; authors state “Single nucleotide polymorphisms (SNPs) of this gene have been identified to predispose carriers to obesity and related comorbidities” however all the SNPs are not within the gene. For instance, rs17782313 is an intergenic variant, located 188 kb downstream the MC4R gene.
I would paraphrase “Single nucleotide polymorphisms (SNPs) mapped to this gene”/ “Single nucleotide polymorphisms (SNPs) located in/near the MC4R”
6) Line 50; authors state “located in different areas of the gene”. Not all are located in the gene. (rs17782313 is an intergenic variant, located 188 kb downstream the MC4R gene). I would give a summary of the SNPs, location and the functional significance to give a clear idea to the readers.
7) Do you have any reference to support “they are located in different areas of the gene where essential transcription promoter elements are located?”. What about the downstream variant rs17782313? It would be more appropriate to say they may have regulatory functions.
8) Line 65; sentence looks incomplete. “in accordance with the principles of the Declarationof Helsinki, and signed informed….?”
9) 12-hour fasting is not relevant for genetic studies, but for lipid biomarkers. Delete 12 hour from genetic and mention under, glucose, lipids, apolipoprotein A1 (ApoA1), apolipoprotein B…..
10) What is the reference chart used for blood pressure Z-score determination?
11) Systolic blood pressure in not a biochemical parameter. I think it fits better with anthropometric biomarkers after changing the subtitle “Anthropometric and clinical parameters”.
12) In table 4; It’s good to indicted significant p-values boldface.
13) In the table 4 caption; define abbreviations; Cys-oC and ApoA1
14) line 280-281; these associations between metabolic biomarkers and SNPs can also be explained by the regulatory functions of non-coding areas in/close to a gene, which can affect protein levels without affecting the structure, hence more relevant in this context. Protein folding, and 3D structure are more relevant in case of SNPs in coding regions.
Author Response
Authors’ Response to Reviewers Comments
Manuscript ID: children-2470749
Title: Association of the rs17782313, rs17773430 and rs34114122 polymorphisms
of MC4R gene with obesity comorbidities in a Spanish paediatric cohort
Dear editor,
Thank you for your kind response on June 20 regarding our manuscript allowing us the opportunity to submit a revised version. On behalf of all the Authors we are very grateful for the kind comments provided by the editor and each of the external reviewers which have helped us to improve the clarity of the manuscript. According to the suggestions, we have incorporated the changes proposed (highlighted in red in the text), incorporated the Table proposed by Reviewer 1, the changes proposed by Reviewer 2 and thoroughly revised the manuscript. We hope that the modifications made will help you for considering the publication in Children. The final version is enclosed and point-by-point responses to the comments are listed below.
Editor Comments
(I) Please check that all references are relevant to the contents of the
manuscript.
Response: It is checked
(II) Any revisions to the manuscript should be highlighted, such that any
changes can be easily reviewed by editors and reviewers.
Response: It is done
(III) Please provide a cover letter to explain, point by point, the details
of the revisions to the manuscript and your responses to the referees’
comments.
Response: It is done
(IV) If you found it impossible to address certain comments in the review
reports, please include an explanation in your appeal.
Response: It is done
(V) The revised version will be sent to the editors and reviewers.
Response: It is done

Round 2
Reviewer 1 Report
In the author's response, there is no answer to my review!
I kindly ask the authors to respond to the previous review.
Author Response
1. We advise you to include a table with basic data about the subjects (age, body weight, height, gender, etc.)
Response: In accordance with your suggestions, these data have been included in Table 1 (Page 4).
2. Furthermore, it would be interested to see how many subjects meet the criteria of metabolic syndrome (see paper https://www.mdpi.com/2227-9067/9/2/204) and the possible association of MS with the mentioned polymorphisms of MC4R gene.
Response: None of the children studied met all the criteria for metabolic syndrome.
Reviewer 2 Report
I have no comments.
The English Language is fine, only minor editing is required.
Author Response
Thank you for your valuable revision